# The Impact of COVID-19 Pandemic on Children with Pulmonary Arterial Hypertension. Parental Anxiety and Attitudes. Follow-Up Data from the Polish Registry of Pulmonary Hypertension (BNP-PL)

**DOI:** 10.3390/jcm10081640

**Published:** 2021-04-12

**Authors:** Joanna Kwiatkowska, Jaroslaw Meyer-Szary, Anna Mazurek-Kula, Malgorzata Zuk, Anna Migdal, Jacek Kusa, Elzbieta Skiba, Karolina Zygielo, Kinga Przetocka, Zbigniew Kordon, Pawel Banaszak, Agata Michalczyk, Alina Rzeznik-Bieniaszewska, Rafal Surmacz, Waldemar Bobkowski, Barbara Wojcicka-Urbanska, Bozena Werner, Joanna Pluzanska, Katarzyna Ostrowska, Magdalena Bazgier, Grzegorz Kopec

**Affiliations:** 1Department of Paediatric Cardiology and Congenital Heart Defect, Medical University of Gdansk, 80-210 Gdansk, Poland; jmeyerszary@gumed.edu.pl (J.M.-S.); magda.bazgier@gmail.com (M.B.); 2Cardiology Department, Polish Mother’s Memorial Hospital-Research Institute, 93-338 Lodz, Poland; qla@op.pl (A.M.-K.); asia.pluzanska@yahoo.com (J.P.); kostrowska9@wp.pl (K.O.); 3The Children’s Memorial Health Institute, 04-730 Warsaw, Poland; zukmala@gmail.com (M.Z.); a.migdal@op.pl (A.M.); 4Regional Specialist Hospital-Research and Development Centre in Wroclaw, Paediatric Cardiology Department, 51-124 Wroclaw, Poland; jkusa@poczta.onet.pl (J.K.); elzskiba@wp.pl (E.S.); kzygielo@gmail.com (K.Z.); 5Department of Paediatric Cardiology, Medical College, Jagiellonian University, 30-663 Krakow, Poland; kprzetocka@usdk.pl (K.P.); zbykord@op.pl (Z.K.); 6Silesian Centre for Heart Diseases, Department of Congenital Heart Disease and Paediatric Cardiology, 41-800 Zabrze, Poland; rhplus@op.pl (P.B.); agat.k.michalczyk@gmail.com (A.M.); 7Department of Paediatric Cardiology, Poznan University of Medical Sciences, 60-572 Poznan, Poland; alina.bieniaszewska@gmail.com (A.R.-B.); rsurmacz@mp.pl (R.S.); wbobk@mp.pl (W.B.); 8Department of Paediatric Cardiology and General Paediatrics, Medical University of Warsaw, 02-091 Warsaw, Poland; barbara.wojcicka@wum.edu.pl (B.W.-U.); bozena.werner@wum.edu.pl (B.W.); 9Pulmonary Circulation Centre, Department of Cardiac and Vascular Diseases, Faculty of Medicine, Medical College Jagiellonian University, John Paul II Hospital in Krakow, u. Pradnicka 80, 31-202 Krakow, Poland; grzegorzkrakow1@gmail.com

**Keywords:** coronavirus, COVID-19, pandemic, pulmonary hypertension, pediatric registry, psychosocial

## Abstract

The COVID-19 pandemic has impacted healthcare systems worldwide. Little is known about the impact of the pandemic on medical and psycho-social aspects of children with rare diseases such as pulmonary arterial hypertension and their parents. The study is based on children registered in The Database of Pulmonary Hypertension in the Polish Population and a parent-reported survey deployed during the first 6 months of the pandemic. The questionnaire consisted of six question panels: demographic data, fear of COVID-19, General Anxiety Disorder-7 (GAD-7), social impact of pandemic, patients’ medical status, and alarming symptoms (appearance or exacerbation). Out of 80 children registered, we collected 58 responses (72.5% response rate). Responders (parents) were mostly female (*n* = 55; 94.8%) at a mean age of 40.6 ± 6.9 years. Patients (children) were both females (*n* = 32; 55%) and males with a mean age of 10.0 ± 5.1 years. Eleven (19%) children had symptoms of potential disease exacerbation. Eight parents (72.7%) decided for watchful waiting while others contacted their GPs or cardiologists (*n* = 6; 54.5%). Three children had to be hospitalized (27.3%). Most planned hospitalizations (27/48; 56.2%) and out-patient visits (20/35; 57.1%) were cancelled, delayed, or substituted by telehealth services. Among the participating parents, the study shows very high levels of anxiety (*n* = 20; 34.5%) and concern (*n* = 55; 94.8%) and the need for detailed information (52; 89.6%) regarding COVID-19 and medical service preparedness during the pandemic. The COVID-19 pandemic has influenced child healthcare and caused high levels of anxiety among parents.

## 1. Introduction

The COVID-19 pandemic caused by SARS-CoV-2 coronavirus has affected all aspects of life [1] and is a huge challenge for healthcare systems around the world [2]. Although the pediatric population seems to be less affected with predominantly an asymptomatic or a mild course of SARS-CoV-2 infection, there are a few groups of patients with a high risk of a severe or fatal course of coronavirus disease. These include children with: congenital heart defects, chronic lung disease, oncologic disorders, neuromuscular diseases, and congenital and acquired immunodeficiency [3,4,5,6,7]. Pulmonary arterial hypertension is a rare progressive disease (ORPHA:182090) [8] leading to heart failure. Therefore, there is a very high risk of a severe or fatal course of COVID-19 [9]. Children with Pulmonary Arterial Hypertension (PAH) are a unique group of patients treated in tertiary centers with special treatment strategies monitored during frequent hospital and out-patient evaluations [10].

The COVID-19 pandemic has had a detrimental effect on chronic disease care in all healthcare systems [2,11,12]. This is not different for those suffering from PAH [13]. It has also led to high levels of anxiety and depression among various patient groups [9,14,15,16]. The patient’s fear of infection during hospital or out-patient visits has caused deterioration of different chronic conditions due to the cessation of systematic medical care [17]. The Polish healthcare model is hierarchical with a GP as a “gate keeper”. Due to the lockdown during the first wave of the pandemic, GPs’ care was limited to telehealth, and specialist care was significantly limited due to the authorities’ decision. This might have been a source of confusion for some patients and parents.

In Poland, children with PAH are treated based on the current guidelines and AEPC 2019 updated consensus statement [8,10] adjusted to the National Health Fund criteria for reimbursement of PAH-targeted therapies. According to the National Health Fund requirements, each drug handover must be conducted during hospitalization. Unfortunately, the COVID-19 pandemic has also put a heavy strain on the National Health Fund in Poland and the distribution of many drugs including target therapies has become uncertain. It was only after the first wave of the pandemic (during the observation time of the study) that the National Health Fund allowed for other handover strategies (e.g., shipping).

In Poland, due to the COVID-19 pandemic, planned cardiosurgical hospitalizations for children have not been cancelled, but postponed. Moreover, within the study observation time, none of the analyzed patients awaited cardiosurgical procedures.

The real-world impact of the COVID-19 pandemic on children with PAH, to the best of our knowledge, has not been shown. In this cross-sectional study, we aim to investigate its impact on children with PAH and their caregivers, medical care, and doctor–patient communication, in addition to anxiety levels and specific COVID-19-related concerns based on the Polish registry of pulmonary hypertension—The Database of Pulmonary Hypertension in the Polish Population [18,19] (BNP-PL, ClinicalTrials.gov Identifier: NCT03959748). These data should provide useful information to guide cardiac healthcare during the COVID-19 pandemic and beyond it for pediatric PAH patients.

## 2. Materials and Methods

### 2.1. Study Group

We recruited a group of parents of children with pulmonary arterial hypertension, who had been enrolled in the BNP-PL registry, to complete an anonymous survey. The design of the BNP-PL registry and enrolment criteria have been recently presented in detail [10,18,19]. In Poland, the management of children with PAH is centralized in eight reference cardiology centers accredited by the National Health Fund to treat PAH. All of the centers have been participating in our registry: The Database of Pulmonary Hypertension in the Polish Population (Baza Nadciśnienia Płucnego; BNP-PL, https://clinicaltrials.gov/ct2/show/NCT03959748; accessed on 10 March 2021).

To be enrolled in the study, all patients had to undergo diagnostic right heart catheterization (RHC), confirming the PAH diagnosis based on the current guidelines [8,10]: mean pulmonary arterial pressure (mPAP) of 25 mm Hg or more, mean pulmonary arterial wedge pressure (PAWP) or left ventricular end-diastolic pressure (LVEDP) of 15 mm Hg or less, pulmonary vascular resistance index (PVRI) of >3 Wood units (WU), and a clinical diagnosis of PAH according to the clinical investigators’ judgments, guided by generally accepted definitions.

All reference centers participated in the study. No personally identifiable information was collected in the survey. Answering the survey was considered to be an implied consent of participation as approved by the institutional review board committee, based on the fact that completing the survey was voluntary, and all answers were confidential. The protocol of the registry was reviewed and accepted by the Bioethical Committee of Physicians and Dentists Chamber in Krakow (L.dz.OIL/KBL/27/2018).

Eligible children were diagnosed with PAH before 20 March 2020, the day the COVID-19 pandemic was officially announced by the Polish government. The survey continued for one week after the responses ceased and was closed at 23:59, 11 September 2020. Outcomes included demographic variables, change in PAH severity during the lockdown period, communication with a treating physician, compliance to the PAH treatment, the difficulty in getting medications, symptoms of anxiety, and pandemic-related concerns among caregivers and patients.

The questionnaire was developed by two investigators and it contains questions that can be answered by an individual parent. Patients’ parents were invited by email. As half of the patients were younger than 10 years old, we found it infeasible for methodological reasons to directly query the minors. The questionnaire consisted of six question panels: Demographic data, Fear of COVID-19, General Anxiety Disorder-7 (GAD-7), Social impact of pandemic, Patients’ medical status, and Alarming symptoms (appearance or exacerbation). If the response was positive, the patient was requested to report their duration. In the fourth panel, we asked patients who had experienced alarming symptoms about any medical contact they had (in person or by phone). Within that set, they were asked about the fear associated with the medical contact caused by the COVID-19 pandemic, the first medical contact, and the impact that the COVID-19 pandemic had on the patients’ decisions and lives. Similar questions were also asked to the parents of children who did not report any alarming symptoms in a hypothetical setting. Additionally, each parent was asked if the child had been diagnosed with COVID-19. A five-level Likert scale (one—strongly disagree, five—strongly agree) was used when appropriate to obtain a uniform question and answer style.

Anxiety was measured by GAD-7, a standard seven-item scale to measure the severity of anxiety. It can objectively determine the severity of initial symptoms and monitor symptom changes and effect of treatment over time. Response to each of seven questions is scored 0–3 points for a total of 0–21 score. It is interpreted as a total of more than 5—mild anxiety, 10—moderate anxiety, and 15—severe anxiety. A total of more than 5, 10, and 15 was interpreted as mild, moderate, or severe anxiety, respectively. Moreover, it was also dichotomized around score 10 to produce None-to-Mild and Moderate-to-Severe classes.

Similarly, several groups of Likert items were summarized as synthetic variables. These sums were then analyzed as continuous variables but also categorized into None, Mild, Moderate, and Severe based on predefined thresholds that were chosen based on the data distribution and to best differentiate the subgroups. “Concerns” (eight items) had thresholds set at 26, 31, and 36. The “Information need” variable (based on 4 items) had thresholds set at 15, 17, and 19. These variables were also dichotomized to aid small-number analysis.

### 2.2. Statistical Methods

Categorical variables were presented as numbers and percentages, and continuous variables as medians and interquartile ranges or means and standard deviations, where appropriate, regarding the data distribution as tested by the Kolmogorov–Smirnov test. The statistical analysis was performed with the use of Wizard 2, Evan Miller. Comparisons between groups for categorical variables were assessed using the chi-square test (Fisher). The Mann–Whitney test was used, and the continuous variable distribution was not found to be normal. Univariate logistic regression was used to determine the association between the different studied variables and the difference in frequency. The statistical significance threshold of the obtained results was set at the *p* < 0.05 level.

## 3. Results

### 3.1. Study Group

On 20 March 2020, the day when the COVID-19 pandemic was officially announced by the Polish government, 80 children with PAH were registered in our pediatric database BNP-PL. A total of 62 parents sent the survey back and 58 of the responses were considered complete and eligible for further analysis (72.5% response rate). Responders were predominantly female, totaling 55 (94.8%) with a mean age of 40.6 ± 6.9 years. Forty-eight (82.7%) families had more than one child. Baseline characteristics of the survey participants (parents) are shown in Table 1.

All children were of Caucasian origin. The mean age was 10.0 ± 5.1 years, with an equal proportion of females, 32 (55.2%), and males, 26 (44.8%). According to the previously established registry database, 32 (55.2%) studied children had PAH related to congenital heart disease (CHD-PAH) and 26 (44.8%) idiopathic PAH (IPAH). Various genetic syndromes were diagnosed in 26 (44.8%) children, most frequently trisomy 21 in 13 (22.4%). Among non-syndromic children, nine (15.5%) had concomitant diseases: epilepsies—three (5.2%), hydronephrosis—three (5.2%), cerebral palsy—two (3.5%), and kidney agenesis—one (1.7%). Additionally, three (5.2%) reported allergy and asthma. All patients were treated with target PAH therapy according to the AHA and AEPC guidelines. In the pre-pandemic period, 31 (53.4%) of children attended nurseries, kindergartens, and schools. The baseline characteristics of these children are listed in Table 1.

### 3.2. Medical Assessment

Until the end of the survey, no patient was SARS-CoV-2-positive. Since the lockdown, 11 (19%) patients suffered the occurrence of a significant exacerbation of the previously existing symptom or multiple symptoms. This included: worsening of exercise tolerance in 10 (17.2%), exertional dyspnea in two (3.4%), syncope or pre-syncope in two (3.4%), cyanosis in one (1.7%), and epistaxis in one (1.7%). No patient reported peripheral edema or ascites (increased abdominal circumference). Consequently, 47 patients (81%) were stable over the course of our study and did not seek medical attention beyond routine visits and drug pick-up.

Most of the parents were well informed and prepared in case of the symptom occurrence. Ten (91%) of those who suffered symptomatic exacerbation said they knew where to report their symptoms, and 38 (81%) reported they would know where to seek help if the symptoms occurred. In the symptomatic group, on the behalf of their children, four (36.4%) parents reported consulting their general practitioner (GP) by phone, five (45.5%) reported consulting their cardiologist by telephone, and one (9.1%) reported visiting their GP in the office. None of the children were seen by their cardiologist in the office and three (27.3%) had to be hospitalized as an emergency case. The reasons for hospitalization were heart failure aggravation and the need for medication adjustments leading to hospitalizations longer than 1 day in each individual situation. In the case of eight (72.7%) children, it was decided to carry out watchful waiting. This section allowed multiple answers and the results are shown on Figure 1. Seven (63.6%) reported that the physician they contacted resolved the problem: in two (50%) cases, it was a GP teleconsultation, and there were three (60%) cardiologist teleconsultations and three (100%) hospitalizations; in seven (63.6%) cases, watchful waiting was sufficient. In the other group (no new symptoms), in case of new symptoms, 14 (29.8%) would try consulting their general practitioner (GP) by telephone, 34 (72.3.%) would try consulting their cardiologist by telephone, four (8.5%) would try visiting their GP in the office, 10 (21.3%) would try visiting their cardiologist in the office, 15 (31.9%) would need to be hospitalized, and six (12.8%) would try watchful waiting. There was no statistically significant difference in how the two groups answered except for watchful waiting being more common in Scheme 0.

The COVID-19 pandemic has clearly influenced the healthcare-seeking behavior in parents of children with PAH. The pandemic has caused a fear of the contact with the health service in the majority of patients, 40 (69%, *p* = 0.004), and has or would have influenced the decisions regarding such contacts in 31 responders (53.4%), most typically leading to a delayed contact in 31 responders (53.4%). The numbers above are based on a pooled sample but there was no statistical difference in how the symptomatic and stable groups responded to the above questions.

As a result of healthcare reorganization during the lockdown, both visits and hospitalizations were postponed or substituted by teleconsultation. With regard to 48 (82.8%) patients during the study period, there was to be the planned hospitalization related to the National Health Fund requirements. Only 21 (46.8%) children were hospitalized as planned, 11 (22.9%) received a new date, two (4.2%) were awaiting a new date, and the rest had a teleconsultation and picked up the medication personally without taking the child to the hospital, 12 (25%), or received it by mail, two (4.2%).

Similarly, 35 (60.3%) patients were awaiting a visit to a specialist clinic. As a consequence of the lockdown, only 15 (42.9%) visits took place as planned, five (14.3%) were substituted by teleconsultation, seven (20%) were rescheduled and the date is already confirmed, and eight (22.9%) were cancelled without specifying a new date of visit.

### 3.3. Social

Before the pandemic, 31 (53.4%) children attended school or kindergarten. After partial loosening of the restrictions, when kindergartens (but not schools) were reopened, only three (33.3%) parents decided that their children would return to kindergartens. Out of the remaining school-aged children, only half of the parents declared they would send their children back to school once they were reopened. Most parents of those attending (26 out of 31, 83.9%) reported being moderately to extremely worried about children returning to school or kindergarten.

Regarding plans for summer holidays, 33 (57%) admitted they will spend them this year differently than before. The majority changed holiday plans due to a fear of COVID-19, 31 (94%), due to the illness of the child/children, eight (24%), and, finally, for financial reasons, five (15%), mostly related to COVID-19, four (12%). The parents declared they would visit their family, 15 (26%), or go on a domestic trip, 15 (26%), and none would go abroad; alternatively, children would go to a summer camp, two (3,4%), or to a rehabilitation facility (3,4%), but the majority would stay home, 32 (55%).

### 3.4. Psychological

The pandemic has caused parents to be concerned about their child’s health more than usually, 55 (94.8%). Specifically, many parents were concerned that COVID-19 can cause complications in their child, 56 (96.6%), including death, 46 (79.3%). They were also afraid of healthcare system organization changes, 35 (60.3%), such as cancellations of planned follow-up visits, leading to unexpected complications, 40 (69.0%), or consequences of the cancellation of planned procedures and surgeries, 35 (60.3%), or simply that they will not receive the information they need, 20 (34.5%). Additionally, they were also afraid of their child’s return to kindergartens or schools, 39 (67.2%). A more detailed breakdown of the data is shown in Figure 2. These items were summarized as “Concerns” for further analysis. This synthetic variable had a uniform distribution with the median score of 30 (range 22–40) and was segregated into categories as shown in Figure 3.

The vast majority of parents expressed a need for more information. Fifty-two (89.6%) parents said they would feel more secure if they knew when the hospitals and out-patient clinics would be reopened and 51 (87.9%) wanted to know when interventions and surgeries would resume. Fifty-six (96.5%) parents wanted to know clearly how to protect their children from the SARS-CoV-2 infection. Fifty-four (93.1%) also wanted to know if children with heart problems such as PAH are actually in the high-risk group if they were diagnosed with COVID-19. These items were summarized as “Information need”. This synthetic variable had no specific distribution with a median score of 18 (range 10–20) and mode of 20 (maximum) and segregated into categories as shown in Figure 3. GAD-7 total scores had a normal distribution and a mean of 9.2 (± 5.5). Twenty (34.5%) responders qualified as moderate-to-severe anxiety (total score > 10) and the detailed categorization is shown in Figure 3.

We found no correlation between parental anxiety, “Concerns”, or “Information need” when four categories were applied, as shown in Figure 3. Using the dichotomous division into None-to-Mild and Moderate-to-Severe categories, we found a correlation between “Concern” and “Information expectation” (*p* = 0.024), a discrepancy likely related to a weak effect size and a small sample size. GAD-7 was still not correlated with any of the two other items. The parental age had no impact on the primary outcomes: GAD-7 (*p* = 0.424), “Concerns” (*p* = 0.150), and “Information need” (0.188). Similarly, the patients’ age also had no impact on: GAD-7 (*p* = 0.439), “Concerns” (*p* = 0.659), or “Information need” (0.316). There was also no correlation with the severity of anxiety, COVID-19-related “Concerns” or “Information need”, and symptomatic aggravation of PAH (*p* = 0.362, 0.215, 0.494, respectively).

While parental anxiety had no influence on the actual or declared healthcare-seeking behavior, the other two dimensions actually did. Thirteen (65.0%) moderately to severely anxious parents, as compared to 18 (47.4%) less anxious parents, reported the pandemic did or would influence their decisions (*p* = 0.201). Similarly, 12 (60.0%) more anxious parents delayed going to the doctor compared to 19 (50%) less anxious parents (*p* = 0.440). Conversely, those having more concerns (moderate-to-severe) claimed the pandemic did or would influence seeking medical attention in 19 (70.4%) vs. 12 (38.7%) of the cases (*p* = 0.016) or delayed seeking help in 15 (55.6%) vs. 16 (51.6%) of the respondents (*p* = 0.764). Similarly, those expecting more information (severe-to-moderate) admitted the pandemic had affected their decisions with 26 (63.4%) vs. five (29.4%, *p* = 0.018) or delayed it with 22 (53.7%) vs. 9 (52.9%, *p* = 0.960). There was no significant difference in the level of fear of contact with the healthcare service and any of the psychological dimensions measured. Interestingly the change in holiday plans was not correlated with the GAD-7 category (*p* = 0.173), “Concerns” (*p* = 0.376), or “Information need” (*p* = 0.799).

## 4. Discussion

As the COVID-19 pandemic unfolds, we are all learning ways to contain SARS-CoV-2 and its effects on children struggling with chronic illnesses [20,21]. The focus of this study is PAH, a rare disease of multiple etiologies. Given the heterogeneity implicit in the pediatric PAH population, predicting the response to COVID-19 is challenging. As the COVID-19 infection has been shown to induce a significant pulmonary, cardiac, and endothelial cell dysfunction, PAH patients may be at particularly high risk for severe complications arising from the COVID-19 infection. Our survey showed a high level of awareness among our patients’ parents in this regard.

Our results demonstrate that during the COVID-19 pandemic, a vast majority of parents of children with PAH responded as being more concerned about their children’s health and complications that COVID-19 can result in. Roughly two thirds had concerns regarding visiting healthcare institutions for fear of contracting COVID-19. Simultaneously, roughly half would let the pandemic situation influence their decisions and they would delay contacting healthcare professionals in this situation, with a substantial proportion ready to withhold acting upon the occurrence of symptoms suggesting primary disease aggravation. These concerns probably simultaneously prompted parents and children to comply better than their healthy peers with protective measures to halt the spread of the disease as none of the patients in the registry had a confirmed SARS-CoV-2 infection up until 11 September 2020.

Simultaneously, dramatic changes in the healthcare delivery in Poland during the COVID-19 pandemic have also affected PAH management and control. Many medical practices, including primary care and pulmonary practices, have limited office visits to urgent patient needs and have largely moved to the telehealth visit mode of communication. Close to half of hospital visits and two thirds of out-patient visits were cancelled or postponed during the lockdown based on the survey, with drugs being delivered by mail or by an indirect contact in a quarter of the cases included. This is concordant with findings of the recent study covering 155 countries conducted by the World Health Organization (WHO Report published 1 June 2020) that have shown that nearly half of the patients with chronic diseases failed to receive their regular medical care and medications since the COVID-19 pandemic began. Our survey results show that roughly two thirds of responders are concerned about the long-term consequences of such a situation in terms of a chronic disease such as PAH in the pediatric population. Changes in medical practice and medication management and use have rapidly altered the immediate PAH management landscape with likely long-term impacts on PAH outcomes also among adults [13] or adult patients with congenital heart diseases (ACHD) [22]. These alterations in healthcare availability, delivery, and utilization have important implications for such a chronic condition as PAH that requires ongoing medical attention.

In our study, only 10% of children reported worsening of symptoms. Approximately one third of symptoms occurred in children on triple-target therapy. In none of the cases did they relate to actual hemodynamic decompensation or result in emergent hospitalization. Yet, careful evaluation and recognition of “red flags” is warranted. Although children appear to suffer a lower burden of illness with COVID-19 infection, they are at risk of developing rare but severe complications including a multisystem inflammatory syndrome unique to children. It is unclear whether underlying conditions such as PAH place children at a greater risk for these rare outcomes.

Although the risk of existing pulmonary hypertension in COVID-19 outcomes received some attention, less discussion has focused on the impact of societal changes. On the presumption that children could be a serious driver of community infection transmission, schools, kindergartens, and nurseries were closed almost ubiquitously around the world to try and halt the potential spread of disease. Most participants reported being worried about their children returning to school or kindergarten despite the pandemic’s effect on the emotional health of the child and themselves, and inter-personal relationships as well as parental productivity. As kindergartens reopened at a later time, only a small proportion of parents decided for their child’s return and only half declared their children would return to school after reopening. The recent pandemic has changed our daily habits, ways we spend time, and travel patterns, significantly impacting family decisions and altering plans.

A stress-related psychosocial impact of the pandemic is evident and complex. Several recent studies documented this effect on patients with chronic diseases [14,16]. A global survey from 47 countries reported worsening of the mental health of 80% of adult patients with chronic diseases during COVID-19 [2]. To measure the severity of anxiety, we used the GAD-7 scale [23]. Its validated sensitivity and specificity based on its authors’ data at a cut-off score of 10 are 89% and 82%, respectively, and test–retest reliability is high (ICC = 0.83) [23]. Later, it also proved useful in primary care and mental health settings as a screening tool and a symptom severity measure for the four most common anxiety disorders (generalized anxiety disorder, panic disorder, social phobia, and post-traumatic stress disorder) [24]. It is 70–90% sensitive and 80–90% specific for these disorders. Higher GAD-7 scores correlate with disability and functional impairment (in measures such as work productivity and healthcare utilization) [23,25]. In the pre-pandemic general population, the GAD-7 score mean was 2.7 (3.2) for women and 3.2 (3.5) for men (with the difference being significant), and for the GAD-7 scores above 10, prevalence reached 5%, and above 15, only 1%. This is in stark contrast to our findings where the mean GAD-7 score was 9.2 (± 5.6) and the respective prevalence levels reached 35% and 21%, making the differences clearly significant.

The majority of parents expressed the need for more information, claiming they would feel more secure if they had more precise information with doubts regarding the ability to protect their children and receiving access to medical care. Although we lack scientific data, we have to assume that children with chronic diseases are at risk. Scientific evidence is hopefully yet to come. Meanwhile, we have to do our best to reassure those families that we are ready to answer their concerns and act in case of emergencies.

The study has several strengths. To our best knowledge, this is the first study to assess the medical and psycho-social impact of the COVID-19 pandemic on children with PAH and their families. Although PAH is a rare disease, we managed to collect a fairly good sample size from multiple institutions. Our study also has limitations. By virtue, all survey studies raise concern about response accuracy and general data quality. Moreover, due to the voluntary participation and response rate being close to 75%, there is a risk of selection bias. The subgroup analysis and comparisons were infeasible for several measures due to the small sample size.

## 5. Conclusions

Our study suggests that in the pandemic era, the prevalence of anxiety among guardians of chronically ill children such as PAH, specifically in this case, is at its peak. Parents are concerned about their children’s health as well as the impact the pandemic has on the healthcare system. They frequently reported adjusting their daily life as well as managing their minor health issues according to the pandemic situation with an intention to minimize risks. The long-term impact of the pandemic and the lockdown on anxiety levels and quality of life requires further study.

## 6. Summary

The COVID-19 changes in medical practice and medication management will likely have major impacts on children with PAH, with many already manifesting. These several risk factors will often work in opposite directions at the same time. By definition, in pandemic circumstances, there is no control group; therefore, it is difficult to foretell the ultimate impact these broad management and social changes resulting from the pandemic will have on patients. Moreover, it was only the pandemic that drew the attention to the anxiety level of parents of children with PAH; thus, no pre-pandemic data regarding this problem are available. The exact relationship and impact of the many contributing risk factors (including issues other than PAH health issues) will be difficult to measure. The limitations of the study also include the homogeneity of the population (all patients were Caucasian), the lack of socioeconomic data, and the impossibility to obtain information directly from the patients due to the young age of the children. Despite these limitations, this paper aims to better guide the management of pediatric PAH and indicate research questions that merit study during these unprecedented times. As PAH is a rare disease, the innovative studies that combine data from the European Children PAH Registries will be important to help the scientific community more broadly understand the pandemic’s full impact. COVID-19 has exposed vulnerabilities in the structural and process measures developed to follow up patients, evaluate the disease status and response to therapy, and screen for complications, each of which depends upon frequent physical patient–physician interactions. Given the uncertain path ahead both with regard to the severity and duration of the COVID-19-related care disruptions, there is a vital need to ensure the best possible outcomes for patients. A follow-up study is underway to investigate how the situation is changing regarding COVID-19 prevalence and survival in this patient group, PAH progression, and how the centers adapted their policies to the ongoing pandemic situation.

## Figures and Tables

**Figure 1 jcm-10-01640-f001:**
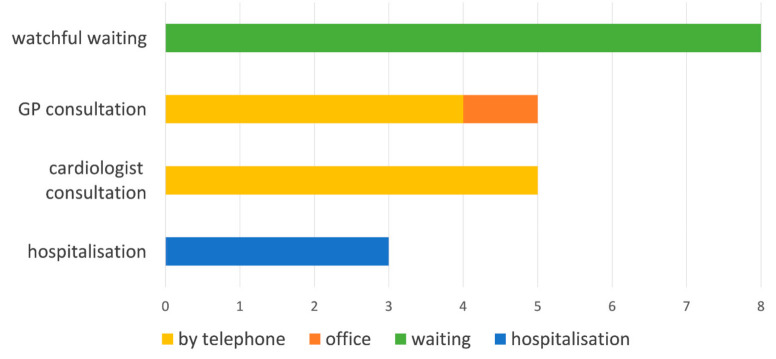
Decision to contact healthcare professional in the symptomatic group.

**Figure 2 jcm-10-01640-f002:**
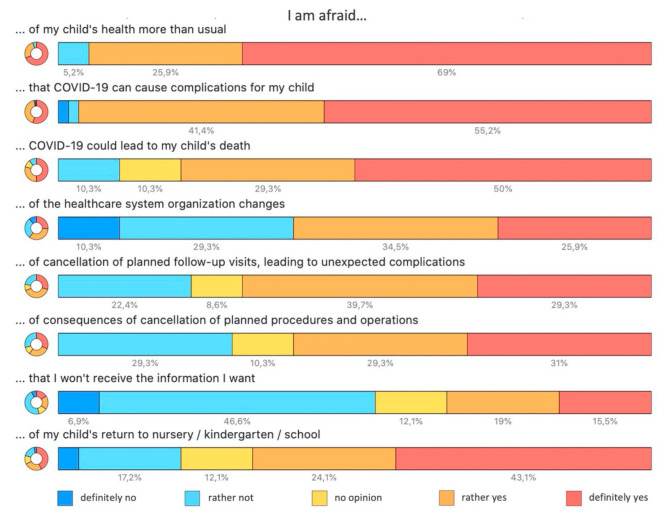
Parental concerns about their PAH (Pulmonary Arterial Hypertension)-suffering children during the COVID-19 pandemic.

**Figure 3 jcm-10-01640-f003:**
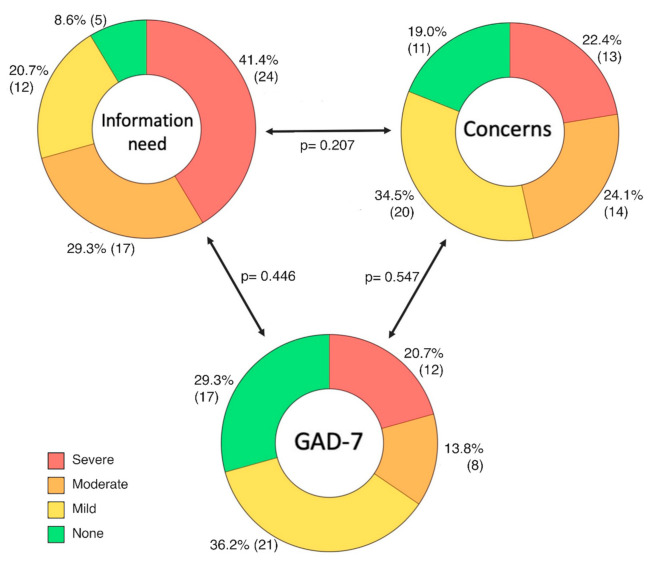
Psychological dimensions analyzed in families with PAH children during the COVID-19 pandemic: information need, concerns related to the pandemic and health, and anxiety as indicated by the GAD-7 questionnaire. Those three dimensions represent the summary of a group of questions and were categorized based on the thresholds specified in the Methods section. Percentages and numbers refer to the adjacent category; *p*-values refer to chi2 test result comparing two dimensions as indicated by arrows.

**Table 1 jcm-10-01640-t001:** Socio-demographic characteristics of the study group.

	N or Mean	% or SD
**Total surveys completed**	58	100%
**Parents’ age (years)**	40.6	6.9
**Parents’ sex**		
female	55	94.8%
male	3	5.2%
**No. of children in the family**		
1	10	17.2%
2	27	46.6%
3	11	19.0%
more	10	17.2%
**Age of patients (years)**	10.0	5.1
**Patients’ sex**		
female	32	55%
male	26	45%
**Education**		
kindergarten	9	15.5%
school	20	34.5%
individual program at school	2	3.4%
individual program at home	11	19%
stayed home (pre-school age)	16	27.6%
**Type of PH**		
CHD-PAH	32	55.%
IPAH	26	4.8%
Genetic syndromes		
Trisomy 21	13	22.4%
Other	13	22.4%
Total	26	44.8%
**Concomitant disease**		
Epilepsy	3	5.2%
Hydronephrosis	3	5.2%
Cerebral palsy	2	3.5%
Kidney agenesis	1	1.7%
Allergy and asthma	3	5.2%
Total	9	15.5%

PH: Pulmonary Hypertension; CHD-PAH: Pulmonary Arterial Hypertension related to congenital heart disease; IPAH: idiopathic Pulmonary Arterial Hypertension.

## Data Availability

No new data were created or analyzed in this study. Data sharing is not applicable to this article.

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
