# Peer review of "The Impact of COVID-19 Pandemic on Children with Pulmonary Arterial Hypertension. Parental Anxiety and Attitudes. Follow-Up Data from the Polish Registry of Pulmonary Hypertension (BNP-PL)"

_jcm, 2021, doi:10.3390/jcm10081640_

Round 1

Reviewer 1 Report

The study of Kwiatkowska and colleagues addresses an important issue in child health, addressing the consequences of Covid-19 pandemic within a vulnerable group of children suffering from pulmonary hypertension.

The strength of the survey is the nationwide sampling of the patients that allows a good overview irrespective of local services that might have had different governmental lock down policies.

The study confirms the psychosocial changes of parents with chronically ill children. As exacerbation of pulmonary hypertension is a life-threatening event it is of great value to analyze the changes of behavior in patients and parents to be prepared for unexpected hospital admissions and adequate alternatives of health care like telecommunications.

  1. Can the authors provide information how many patients with PH/PAH are not included in the registry?
  2. It does not get clear, why the patients themselves have not been analyzed in more detail to distinguish the impact of the parents’ anxieties projected to the children.
  3. Quite a number of patients within the cohort are syndromic – do the authors expect an even higher risk for example due to Trisomy 21?
  4. Data on the patient’s medication are missing what makes it difficult to understand anxiety not to have access to target therapies. Readers not familiar with the polish distribution system of these medication would be confused – it should be mentioned.
  5. As it is mentioned that a high portion of patients has congenital heart defects – it is not clear if a planned hospitalization for heart surgery had to be cancelled. The authors must report on that.
  6. The authors quoted the age of the parents but did not discuss in detail if there is any impact of the age of the parental caregivers to be expected. If so, it should be elaborated.
  7. To better understand the reduced visits at doctors during the pandemic: Did the patients have to contact their known care givers or did that change since the pandemic so to say, why should they not know whom to contact?
  8. Is there a hierarchic system in Poland, so they had to contact GP first or could they have addressed the specialist first – would be a helpful information to judge the analysis.
  9. Figure 1 should be clarified whether the pediatrician is the same as the GP, as otherwise a bar with the GP would be missing. If it is two groups - they should be splitted and displayed within the figure.
  10. Could you clarify the reason for hospitalization! Because that would mean very unstable hemodynamics in the patients or does it mean that they should be seen in a specialized outpatient clinic within a hospital? If so, this must be clear, as internationally hospitalization would mean stay in hospital for more than one day.
  11. Figure 3 is very difficult to understand, the legend should be optimized.
  12. The authors state that there has been no single case with Covid-19 within the study period. As this was only the start of the pandemic a follow up study would be really interesting. If there are still low numbers of PAH patients that worsen considerably, eventually the survey would now be different also including the question about holidays.
  13. Did the centers change their policy of patient care due to the results of the study? Nurse practioners to stay in contact with the families? Psychologists that are available for the patients and the parents…….
  14. Did the registry meanwhile has patients that worsened or even died and if so, did this result in changes of opening doctoral offices or changes in the distribution of medications for rare diseases.

Author Response

Reviewer 1

Comments and Suggestions for Authors

The study of Kwiatkowska and colleagues addresses an important issue in child health, addressing the consequences of Covid-19 pandemic within a vulnerable group of children suffering from pulmonary hypertension.

The strength of the survey is the nationwide sampling of the patients that allows a good overview irrespective of local services that might have had different governmental lock down policies.

The study confirms the psychosocial changes of parents with chronically ill children. As exacerbation of pulmonary hypertension is a life-threatening event it is of great value to analyze the changes of behavior in patients and parents to be prepared for unexpected hospital admissions and adequate alternatives of health care like telecommunications.

  1. Can the authors provide information how many patients with PH/PAH are not included in the registry?

Authors’ response:

Based on the enrolment criteria all alive patients diagnosed with PAH are included in the registry.

In Poland, the management of children with PAH is centralized in eight reference cardiological centres accredited by the National Health Fund to treat PAH. All of the centres have been participating in our registry; The DataBase of Pulmonary Hypertension in the Polish Population (Baza Nadciśnienia Płucnego; BNP-PL, https://clinicaltrials.gov/ct2/show/NCT03959748).

To be enrolled in the study, all patients had to undergo diagnostic right heart catheterization (RHC), confirming the PAH diagnosis based on the current guidelines [8,10]: mean pulmonary arterial pressure (mPAP) of 25 mm Hg or more, mean pulmonary arterial wedge pressure (PAWP) or left ventricular end‐diastolic pressure (LVEDP) of 15 mm Hg or less, pulmonary vascular resistance index (PVRI) of >3 Woods Units (WU) and a clinical diagnosis of PAH according to the clinical investigators’ judgments, guided by generally accepted definitions.

Unfortunately, other groups of PH are not included in the registry, so we do not know the total number of PH children in Poland.

  1. It is not clear why the patients themselves have not been analyzed in more detail to distinguish the impact of the parents’ anxieties projected to the children.

Authors’ response:

With half of the patients being younger than 10-years-old, we found this infeasible for methodological reasons to directly query the minors.

  1. Quite a number of patients within the cohort are syndromic – do the authors expect an even higher risk for example due to Trisomy 21?

Authors’ response:  

Yes, we expected a higher risk of the SARS-COV-2 infection in children with genetic syndromes, especially Trisomy 21. Our survey conducted during the first wave of the pandemic did not confirm this concern.

  1. The data on the patients’ medication is missing which makes it difficult to understand anxiety not to have access to target therapies. Readers not familiar with the polish distribution system of these medication would be confused – it should be mentioned.

Authors’ response:  

In Poland, children with PAH are treated based on the current guidelines and AEPC 2019 updated consensus statement [8,10] adjusted to the National Health Fund criteria for reimbursement of PAH targeted therapies. According to the National Health Fund requirements each drug handover must have be conducted within the hospitalization. Unfortunately, the COVID – 19 pandemic has put a heavy strain on the National Health Fund also in Poland and the distribution of many drugs including target therapies has become uncertain. It was only after the first wave of the pandemic (during the observation time of the study) that the National Health Fund allowed for other handover strategies (e.g. shipping).

  1. As it is mentioned that a high portion of patients has congenital heart defects – it is not clear if a planned hospitalization for heart surgery had to be cancelled. The authors must report on that.

Authors’ response:  

In Poland, due to the Covid-19 pandemic, planned cardiosurgical hospitalizations for children have not been cancelled, but postponed. Moreover, within the study observation time, none of the analysed patients awaited cardiosurgical procedures.

  1. The authors quoted the age of the parents but did not discuss in detail if there is any impact of the age of the parental caregivers to be expected. If so, it should be elaborated.

Authors’ response:

We removed the following sentences: “Those psychological measures were also not correlated to patients’ age or parents’ age (all p > 0.05). There was also no correlation with the severity of anxiety, COVID-19 related concerns or information need and symptomatic aggravation of PAH (p = 0.362, 0.215, 0.494 respectively).“ The statement was replaced with the following: “Parental age had no impact on the primary outcomes: GAD-7 (p=0.424), concerns (p=0.150) and information need (0.188). Similarly, the patients age also had no impact on: GAD-7 (p=0.439), concerns (p=0.659) or information need (0.316).”

  1. To better understand the reduced visits at doctors during the pandemic: Did the patients have to contact their known care givers or did that change since the pandemic so to say, why should they not know whom to contact?

Authors’ response:

The Polish health care model is hierarchical with the GP as a “gate keeper”. Due to the lockdown during the first wave of the pandemic, the GP’s care was limited to telehealth, and the specialist care was significantly limited due to the authority’s decision. This might have been a source of confusion for some patients and parents.

  1. Is there a hierarchic system in Poland, so they had to contact GP first or could they have addressed the specialist first – would be a helpful information to judge the analysis.

Authors’ response:

Yes, the Polish health care model is hierarchical.

  1. Figure 1 should be clarified whether the pediatrician is the same as the GP, as otherwise a bar with the GP would be missing. If it is two groups - they should be splitted and displayed within the figure.

Authors’ response:

According to the nomenclature used in the study, a pediatrician was the same as the GP. Appropriate changes were applied both in the manuscript and Figure 1 replacing the term “pediatrician” with “the GP” for better understanding and clarity.

  1. Could you clarify the reason for hospitalization! Because that would mean very unstable hemodynamics in the patients or does it mean that they should be seen in a specialized outpatient clinic within a hospital? If so, this must be clear, as internationally hospitalization would mean stay in hospital for more than one day.

Authors’ response:

The reason for 3 hospitalizations was the heart failure aggravation and the need for medication adjustments leading to hospitalizations longer than 1 day in each individual situation. The other hospitalizations were associated with drug collection during hospitalization, according to the National Health Fund requirements.

  1. Figure 3 is very difficult to understand, the legend should be optimized.

Authors’ response:

The Figure 3 legend has been modified for better understanding.

  1. The authors state that there has been no single case with Covid-19 within the study period. As this was only the start of the pandemic a follow up study would be really interesting. If there are still low numbers of PAH patients that worsen considerably, eventually the survey would now be different also including the question about holidays.
  2. Did the centers change their policy of patient care due to the results of the study? Nurse practioners to stay in contact with the families? Psychologists that are available for the patients and the parents…….
  3. Did the registry meanwhile has patients that worsened or even died and if so, did this result in changes of opening doctoral offices or changes in the distribution of medications for rare diseases.

Authors’ response:

A follow-up study is underway to investigate how the situation is changing regarding the COVID-19 prevalence and survival in this patient group, PAH progression and how the Centres adapted their policies to the ongoing pandemic situation.

Reviewer 2 Report

Thank you for the opportunity to review the manuscript by Kwiatkowska et al titled “The Impact of COVID-19 Pandemic on Children with Pulmonary Arterial Hypertension. Parental Anxiety and Attitudes. 3 Follow-up Data from the Polish Registry of Pulmonary Hyper- 4 tension (BNP-PL)”. It is a very well written manuscript looking at the children’s psychological status and parental anxiety prevalence among patients with pulmonary hypertension. However, there are a lot of minor grammatic and language errors, some of which I have mentioned.

My comments:

Abstract:

Line 30: Change to “impact of the pandemic on…

Line 34: Don’t capitalize words after comma.

Line 36: Change 72,5 to “72.5”

Line 38: Report as mean age

Introduction:

Line 56: change to “congenital and acquired immunodeficiency”

Line 58: change to an “unique…

Line 80: change cardiological to “cardiology”

Materials and Methods:

Line 128: Rephrase the sentence mentioning logistic regression for more clarity

Results:

Line 138: change to “mean age of”

Did a pre pandemic psychologic status assessed?

How do we know that other disorders apart from PHTN contribute to the psychological stress in these patients and parents?

Line 192: Planning hospitalization or planning to visit a doctor?

Discussion:

Why no data on socioeconomic status collected? This can attribute to anxiety unrelated to the PHTN but related to other changes that could have happened in the setting of a pandemic. Eg: Job loss

All enrolled patients were Caucasian. That is a limitation that should be included
